# Cross-Cultural Adaptation and Evaluation of the Participation and Environment Measure for Children and Youth to the Indian Context—A Mixed-Methods Study

**DOI:** 10.3390/ijerph18041514

**Published:** 2021-02-05

**Authors:** Roopa Srinivasan, Vrushali Kulkarni, Sana Smriti, Rachel Teplicky, Dana Anaby

**Affiliations:** 1Ummeed Child Development Center, Department of Developmental Pediatrics and Occupational Therapy, Mumbai 400011, Maharashtra, India; vrushali.kulkarni@ummeed.org; 2Butterflies Child Development Centre, Hyderabad 500081, Telangana, India; drsanasmriti@gmail.com; 3CanChild Center for Childhood Disability Research, McMaster University, Hamilton, ON L8S 4L8, Canada; teplicr@mcmaster.ca; 4School of Physical and Occupational Therapy, McGill University, Montreal, QC H3G 1Y5, Canada; dana.anaby@mcgill.ca

**Keywords:** rehabilitation, PEM-CY, participation, cultural adaptation, India

## Abstract

Culturally appropriate measures enable knowledge transfer and quality improvement of rehabilitation services in diverse contexts. The Applied Cultural Equivalence Framework (ACEF) was used in a two-phased mixed methods study to adapt and evaluate the Participation and Environment Measure-Children and Youth (PEM-CY) in India. Cognitive interviews with caregivers of children with disabilities (*n* = 15) aged 5–17 years established conceptual, item, semantic, and operational equivalence of the Indian PEM-CY. Construct validity was assessed by comparing PEM-CY scores of children with and without disabilities (*n* = 130) using a case-control design. Cognitive interviews resulted in operational (60.3%), semantic (26.4%), and item-level (13.2%) modifications in the PEM-CY with no changes at the conceptual level. Internal consistency (*n* = 130) was acceptable to excellent (0.61–0.87) on most scales. Test–retest reliability (*n* = 30) was good to excellent (ICC ≥ 0.75, Kappa 0.6–1.0) for most scales. Significant differences in all PEM-CY summary scores were found between children with and without disabilities, except for environmental supports. Children with disabilities had lower scores on frequency and involvement in activities across all settings; their caregivers desired greater change in participation and reported experiencing more environmental barriers across settings. Findings suggest the adapted PEM-CY is a valid and reliable measure for assessing the participation of Indian children.

## 1. Introduction

Participation in daily activities that one wants to and/or is expected to be engaged in is globally recognized as an indicator of health and as one of the most important outcomes of rehabilitation interventions [1,2]. The International Classification of Functioning, Disability, and Health (ICF) framework [3], endorsed by the World Health Organization, emphasizes the role of environmental factors in positively or negatively affecting an individual’s participation. Indeed, previous research demonstrated that participation is a highly individualized, multidimensional concept that is dependent on environmental factors such as the physical (e.g., built environment), social (e.g., family and peer support), cultural (e.g., attitudes and values), and institutional (e.g., availability of program, services, and inclusive policies) aspects of the environment [4,5]. Researchers recommend that the ICF framework and participation-related research should guide the development of measures for children’s participation [6,7,8]. The Participation and Environment Measures (PEM) are examples of such measures that uniquely look at both participation and environmental barriers and facilitators for children and youth with and without disabilities across home, school, and community settings [9,10]. The PEM also assesses key elements of participation; attendance (“being there”) and involvement (“being in the moment”) [5], making them one of the most comprehensive tools available. Specifically, the Participation and Environment Measure—Children and Youth version (PEM-CY), developed in North America, is a psychometrically sound parent-report assessment intended for children aged 5–17-years-old [6]; however, it has yet to be adapted to the unique context of low-resource countries.

Low and middle-income countries such as India are home to 95% of the world’s children with disabilities under the age of five years [11]. Global estimates for older children with disabilities, though unavailable, are likely to follow a similar trend and very little information is available on the participation and well-being of these children [12,13]. While resources and services for children with disabilities have been steadily increasing, the caregivers face several challenges while accessing them. Services infrequently consider psychosocial factors and their influence on a person with a disability or their family [14]. Societal stigma and cultural beliefs may often force caregivers to seek “fixes” and cures for their child’s disability [14]. Low caregiver literacy levels limit access to information about disabilities and available services [15]. Policies and laws that are supportive of children with disabilities are often poorly utilized on account of poor awareness and weak regulatory mechanisms [15]. Cultural preferences and inadequacy of formal supports for interventions prompt caregivers in under-resourced contexts to seek informal sources of supports within the family and the community [13]. The opportunity to leverage these supports for child and caregiver well-being is often missed by providers because of the focus on finding a fix for the disabilities. The availability of a measure like the PEM-CY would provide an opportunity for measuring children’s participation, the impact of the environment on the child’s participation, and evaluating the outcomes of rehabilitation services in low resource contexts.

### Objectives

To culturally adapt and evaluate PEM-CY to the Indian context using the Applied Cultural Equivalence Framework (ACEF). Specifically, we aimed to (1) adapt the content of the PEM-CY and its administration to the Indian culture (phase 1) and (2) examine the psychometric properties of the adapted version in terms of reliability (internal consistency and test-retest) and construct validity (phase 2).

## 2. Materials and Methods

### 2.1. Study Design

A two-phase mixed-method design [16] was used to culturally adapt and test the English and Hindi PEM-CY among parents living in India. This process was guided by the five criteria outlined in the Applied Cultural Equivalence Framework (ACEF) [17,18,19] as shown in Table 1. In phase 1, the conceptual, item, semantic, and operational criteria of the ACEF were used to adapt the measure to the local context after considering the influence of the local culture, language, and interpretation of the construct. The inclusion of operational criterion is unique to this framework, which helps to evaluate not just the content of the instrument, but also its administration. This is critical considering the diversity in educational background and familiarity with self-administered instruments in the study setting. The ACEF framework has been used previously for cultural adaptation for various participation instruments [17].

This study included two phases, as shown in Figure 1.

### 2.2. Procedure

The study was conducted at a large, not-for-profit organization serving children with disabilities between 0 and 18 years of age, during September 2018–July 2019. Ethics committee approval was obtained from an Institutional Review Board (Approval ID 01/2019) on 20th of April, 2019. Informed consent was obtained from the caregivers in both parts of the study.

Phase 1: PEM-CY was forward and backward translated in Hindi and English respectively by research assistants who were proficient in both languages and unfamiliar with the measure as per guidelines suggested by Beaton and colleagues [21]. A conceptual strategy [22] was used for the translation, where the significance of the items and instructions was preserved, rather than aiming for a direct text translation. Descriptive phrases were used when an equivalent word in Hindi was unavailable. Widely understood English words were retained where applicable. The backward translated English version of the PEM-CY was compared with the original PEM-CY to verify the accuracy and consistency of translation by the authors and developers of the measure.

Two rounds of cognitive interviews were used to establish the conceptual, item level, operational, and semantic equivalence of the translated version of the PEM-CY. To ensure diversity of perspectives, an important principle of cognitive interviews, 15 caregivers (10 for the Hindi version and 5 for the English version) of children with disabilities between the ages of 5–17 years were purposefully recruited, as recommended by Peterson et al. [23]. Caregivers were asked to choose either the original English PEM-CY or the translated Hindi version based on their comfort with reading, understanding, and completing a questionnaire in either of these languages. An open-ended interview guide was developed (based on the ACEF criteria) to conduct semi structured cognitive interviews using the Think Aloud and Verbal Probes approach in the first and second round of cognitive interviews respectively [23,24,25]. In the Think Aloud approach, the caregivers are encouraged to verbalize their thought processes with the interviewer performing the role of listening and using minimal prompts to avoid interruption in thinking (e.g., what made you choose this response?) [23]. In the Verbal Probes approach, specific pre decided probes are used immediately after the caregivers responded to the item (e.g., what did you understand by the example given here?) to confirm that the modifications made were helpful and enquire if additional modifications were needed [23]. The second round involved five caregivers who had not completed high school education. Modifications recommended by this group of caregivers (round 2) were hypothesized to be acceptable to caregivers with higher levels of education.

Phase 2 of the study assessed the psychometric properties of the PEM-CY after modifications made in Phase 1. The scores generated by the modified PEM-CY was used to assess our hypothesis regarding significant differences in participation between children with and without disabilities. A case-control study design was employed including caregivers of children with (*n* = 65) and without disabilities (n = 65) who were matched by the respondents’ educational level and child’s age and sex [26]. Caregivers of children were included in the study if (1) they had a child with or without disabilities between the ages of 5–17 years; (2) they could read and understand English or Hindi; (3) they lived in Mumbai; and (4) provided written consent to participate in the study. Internal consistency was examined in this cohort (*n* = 130) and test–retest reliability was assessed among a subsample of 30 parents at two-time points with a delay of 2–4 weeks [6,27]. Image 1 illustrates the procedure of the study.

Caregivers of children with disabilities in both phases (who participated in the second round of cognitive interviews and Phase 2 of the study) rated the feasibility and understandability of the instrument. Additionally, in phase 2, research assistants documented the type of assistance needed by caregivers of children with and without disabilities in completing the Indian-PEM-CY using the following criteria: redirecting caregivers to survey guidelines, prompts that included reading assistance and reminders to think about their everyday experiences and discussions to help caregivers connect their everyday experiences with the items in the measure (Appendix A).

All data was deidentified in both phases with restricted access to the research team. In Phase 1, the audio/video recording of the cognitive interviews was used to transcribe and translate the interviews. The transcripts were saved securely in a password-protected folder and thumb drive. In Phase 2, data from the completed PEM-CY paper forms was entered on MS Excel software (Microsoft Excel Macro Enabled Worksheet, Mumbai, Maharashtra).

### 2.3. Measurements

The PEM-CY, a caregiver-report measure, was used to assess the participation patterns of children and youth, aged 5–17, and the impact of the environment of their participation [6]. It includes 25 items focused on participation in broad types of activities at home (10 items), school (5 items), and community (10 items) settings. For each item, the caregiver reports on three dimensions of the child’s participation: (1) frequency (8-point scale, from never (0) to daily (7)); (2) level of involvement (5-point scale, from minimally involved (1) to very involved (5)); and (3) the caregiver’s desire for change in the child’s participation (yes or no; if yes, the parent can select whether he or she desires a change in frequency, level of involvement, and/or broader variety). For each setting, the parent also reports on whether various environmental features or resources impact their child’s participation. There are 12 environmental items in the home setting, 17 for school, and 16 for the community. The PEM-CY has moderate to very good internal consistency (0.59–0.91) and moderate to good test–retest reliability for all participation and environment sections when assessed within a 4-week period after the completion of first round (ICC 5 0.58–0.95) [6]. The validity was established as PEM-CY detected a significant effect of disability on child’s participation across all settings and variables [6]. Recommendations provided in the manual were used to calculate summary scores to facilitate comparisons among groups [28]. Four mean scores were calculated to illustrate participation patterns in each setting: number of activities participated (in percentages), frequency (mean score ranging from 1 to 7), involvement (mean score ranging from 1 to 5), and desire for change (number of activities in which parented wanted to see changed, presented in percentages). Two additional scores per setting were calculated to describe the number of the environment supports and barriers, in percentages. Thus, six average group summary scores in each of the three settings (overall 18 scores) were derived.

Caregiver perceptions about the feasibility of the use of the instrument, understandability, relevance of items, and examples for home, school, and community were measured using several Visual Analogue Scales (VAS) [29,30]. The VAS included a 10 cm scale with 0 being “Not understandable and irrelevant” and 10 being “understandable and relevant”. Overall, three VAS mean scores were generated for each of the three settings, and one overall score to assess caregivers’ perception about the relevance and feasibility of use of the measure resulting in a total of 10 scores ranging from 0 to 10. Further, caregivers participating in phase 2 completed 4 questions related to the overall relevance, whether the PEM-CY should be used in the intervention, its understandability, and time taken to complete the measure with a simple “Yes” or “No” response. These were reported as the absolute number of responses and percentages.

Child and family characteristics, such as the caregiver’s relationship with the child, age, and education levels, were collected using a standard demographic questionnaire. Information about the child including their age, gender, diagnosis, and functional limitations was also reported by the caregivers. Parents reported the diagnoses (up to 3) and functional limitations using a scale [31]. The diagnoses involved developmental disabilities like Autism Spectrum Disorder, Learning Disability, Developmental Delay, and Intellectual Disability and health conditions like asthma, cardiac problems, and epilepsy. The diagnoses for children with disabilities receiving services at our center was made by Developmental Pediatricians using the Diagnostic and Statistical Manual (DSM) V criteria [32]. When caregivers could not report the diagnoses, it was retrieved from patient records. The severity of the child’s condition in terms of the number of functional limitations was reported by the caregivers by indicating whether a functional skill was “not a problem”, a “little problem” or a “big problem” using a checklist of 11 functional areas. The number of functional issues was tallied. This checklist was found effective in explaining levels of participation among children and youth across different disabilities [6].

### 2.4. Data Analysis

Phase 1—Cultural adaptation of the PEM-CY to the Indian context (ACEF criteria for content and administration):

To analyze information gathered by the sequential rounds of cognitive interview, a deductive coding approach [33] was used to organize findings according to the first four criteria of the ACEF for all three settings home, school, and community section. In each round of cognitive interviews, similar modifications were grouped and counted as one, to avoid duplication in counting. Two of the authors independently reviewed the coding and the appropriateness of their listing under each of the ACEF criteria and any discrepancies were resolved by discussion. A summary report of these findings across the 15 interviews was sent to the codevelopers of the original measure. The proposed changes were reviewed, discussed, and any divergence of opinion reconciled by consensus of all the authors.

Phase 2—Testing the psychometric properties of the adapted PEM-CY (ACEF criteria measurement):

To examine construct validity, we assessed the extent to which the scores of the adapted PEM-CY was consistent with our hypothesis about difference in participation, involvement, change desired, environmental supports, and barriers between children with and without disabilities [34]. An unpaired *t*-test was used if the data was normally distributed, whereas the Mann–Whitney U test was used if the data failed the “Normality” test, as determined by the Shapiro–Wilks test [35]. A *p*-value of less than or equal to 0.05 was considered as a cut-off for the failure of the normality test. Effect sizes were calculated using Cohen’s d where *d* = 0.2 is considered a small effect, *d* = 0.5 is medium, and d = 0.8 is large [36]. Reliability (i.e., internal consistency and test–retest reliability) was assessed using the Cronbach Alpha, Intraclass Correlation (ICC), and the Kappa Agreement test, respectively. The test–retest pairs for each individual Likert scale item in each of the three settings were analyzed using Intraclass Correlation [37] and when the scores were dichotomous, the simple Kappa Agreement was used [38]. Kappa scores ranging from 0 to 1 were interpreted using Landis and Koch guidelines [39]. ICC values less than 0.5 are indicative of poor reliability, values between 0.5 and 0.75 indicate moderate reliability, values between 0.75 and 0.9 indicate good reliability, and values greater than 0.90 indicate excellent reliability [40]. Values of *p* < 0.05 were used as the cut-off for statistical significance. IBM PSPP version 1.0.1 was used to analyze data recorded in MS EXCEL (Microsoft Excel Macro Enabled Worksheet, Mumbai, Maharashtra).

## 3. Results

### 3.1. Phase 1—Caregiver and Child Characteristics

The sociodemographic profile of the 10 caregivers in the 1st round (Hindi and English) and the 5 caregivers from the 2nd round of cognitive interviews along with information about their children is presented in Table 2. Thirteen mothers and two fathers of children with disabilities participated in this study. All caregivers resided in the Mumbai Metropolitan Region.

#### Modifications in English and Hindi PEM-CY

The majority of the changes were operational in nature (60.3%), followed by semantic (26.4%) and item-level changes (13.2%). No conceptual changes were required in either round of the cognitive interviews. As the second round of cognitive interviews was conducted with caregivers who completed the Hindi PEM-CY there were no modifications that were unique to the English PEM-CY after this round. There were fewer item-level modifications needed in the second round (16.6% vs. 9.8%), similar operational modifications in both rounds (61.6% vs. 59%), and a higher number of semantic changes that were needed in round 2 (21.6% vs. 31.1%). The lower education levels of the caregivers participating in the 2nd round of cognitive interviews may have led to the increased numbers of semantic and operational changes in the second round of the cognitive interviews overall. Overall, the modifications proposed remained the same in both rounds of cognitive interviews.

While the caregivers did not perceive the need for any changes at a concept level, 13 out of 15 caregivers needed reminders that “involvement” is about attention, concentration, and emotional engagement in activities and not about “independence”. This was especially seen in the home section of the PEM-CY. Once reminders were given parents were able to complete and relate to these questions and therefore this was listed under operational modifications. Caregivers tended to focus on the child’s difficulties and impairment rather than on environmental barriers or supports. They required an additional set of instructions and examples to understand this concept.

For the item-related modifications, contextually relevant examples made the items easier for the caregivers to understand. For example, adding “mobile” to the item “Computer and Video games” in the home setting. Caregivers could relate to cultural programs organized in their residential area or apartment during religious festivals and public holidays and therefore we modified the item “Programs in Community” to “Programs organized by the apartment/ building”. The word “etcetera” was added to encourage caregivers to think more broadly about the type of activities that could be considered within an item as, at times, they felt constrained by the examples provided. Fewer item-level modifications were needed in Round 2 of cognitive interviews.

In terms of semantic modifications, two types of changes were made in both the English and Hindi PEM-CY. In the Hindi PEM-CY, English words like “puzzles”, “craft”, and “class” were retained as they were more commonly used and better understood. Colloquial words and descriptive phrases in Hindi were preferred over a literal and poorly understood Hindi translation; for example, “support” was replaced by “help”, phrases were used to explain words or phrases such as “community”, “organized physical activities”, and “unstructured physical activities”. For example, “community” was replaced by “locality and nearby area”, and “organized physical activities” was modified to “physical sports organized in a private class”. All the semantic modifications made to the Hindi PEM-CY after the second round of cognitive interviews were replicated in the English PEM-CY. Additionally, in the English PEM-CY, words like “field trips” and “dress up games” were replaced by “picnic” and “dress up with saree or dupatta” respectively. The words “Survey Instruction” was replaced by “Survey Guide”, “independence” was replaced by “child’s abilities”, and “cognitive demands” of the activity by “brain/thinking demands”.

Operational modifications were often required; the majority (61.6% in Round 1 and 59% in Round 2) of the changes were operational and involved changes to the layout and the administration of the questionnaire. The interviewers observed challenges faced (skipping a column) or strategies used by the caregiver (using their finger to keep track of rows) while completing the instrument. Caregivers recommended modifications that involved clarifications of survey instructions, formatting changes, and highlighting key transition points within the PEM CY.

Clarifying Survey Instructions included providing explanations for concepts such as “involvement” and pictorial examples for completing the participation and environment sections. A sample of a completed PEM-CY question with clear instructions for entering responses was added to the survey instructions (see Figure 2). Specific pointers such as “For question B on Involvement” were used instead of the less specific term “important” that was part of the original survey instructions. The stem questions and response options were elaborated and made more self-explanatory after Round 2. For example, the question “What do you do to support your child’s participation?” was modified to “What do you do to support your child’s participation at present?” to make the question easier to understand and respond to. Contextually relatable examples were used for environmental supports that could be made available to enhance the child’s participation (e.g., In the survey guidelines-wheelchair to support mobility was used as an example of environmental support).Reformatting of the questionnaire was also required. Specifically, modifications such as increased font size, color coding, italicizing, underlining, the spacing between items, and columns were used to draw the attention of the caregiver to important instructions and steps in both rounds of interviews.Help with transitions within the PEM-CY sections was also needed. Transition boxes were added between the participation and environment section to alert the caregiver to the change in section.

Results of the VAS questions indicated a relatively high level of understandability and relevance of items and relevance of examples from the modified PEM CY in the community (8.8, 9, and 9.4 out of 10, respectively), followed by the school (8, 8.2, and 8.8 out of 10) and the home setting (6.6, 8.6, and 9.4 out of 10). On a maximum score of 10, the caregivers rated the feasibility of PEM-CY at 8.8 on average.

### 3.2. Phase 2

#### 3.2.1. Caregiver Characteristics (*n* = 130)

Initially, 73 caregivers of children with disabilities (case group) completed the modified PEM-CY questionnaires. This sample was matched with caregivers of children without disabilities (control group). Responses from 65/73 caregivers were complete with less than 20% missing data and could be included in the final data set for analysis. The sociodemographic profile of the case (*n* = 65) and control group (*n* = 65) was included in Table 3. A nearly equal number of caregivers in the case and control group completed the Hindi and the modified English PEM CY. From among the functional issues, most of the caregivers, 66% (43/65), reported issues in 4–9 (Median 5) functional areas. The most common functional issues included difficulty in “paying attention” (78%), “communication” (66%), “managing emotions” (60%), “remembering information” (58%), and “controlling behaviors” (57%) in that order. On the lower end, 25 (38%) reported difficulties in moving around, 16 (25%) in use of hands, 9 (12%) in vision, and 8 (13%) in hearing.

#### 3.2.2. Psychometric Properties

##### Construct Validity of the Adapted PEM-CY

Significant differences, with moderate to large effect sizes, in participation frequency and involvement were found between children with disabilities and their typically developing counterparts across all settings: home, school, and within the community (see Table 4). As expected, children with disabilities participated less often and were less involved in activities at the home, school, and the community. In addition, a significantly greater number of caregivers (with a large effect size) of children with disabilities desired change in participation in all three settings. There was no statistically significant difference in the diversity of activities at home, school, and community. Children with disabilities were like their typically developing counterparts in viewing television, socialization using technology and participation in classroom work. They participated less frequently in indoor games, household chores, preparing for school, socializing with peers at home and school, and in all community-based activities. Caregivers of children with disabilities identified significantly more (with a large effect size) environmental barriers for their children’s participation across settings as compared to caregivers of children without disabilities. Social demands of activities, inadequate money, supplies, services, and information were identified as barriers to participation at home by caregivers of children with disabilities. Cognitive and social demands along with inadequate money, services, supplies, transportation, policies and procedures were considered as barriers to participation in school and community. Differences in the perceived environmental support were evident descriptively (lower number of supports reported among those with disabilities), yet no statistical significance across any of the settings was observed.

##### Reliability of the Adapted PEM-CY

Estimates of internal consistency of items pertaining to all the scores across settings, examined among the entire cohort (*n* = 130), were acceptable to very good (0.61–0.87), as shown in Table 4 The test–retest reliability was examined among a subsample of 30 caregivers who participated; most of them had education levels above graduation (26/30) and earned more than minimum wage (29/30). Estimates of intraclass correlation were greater than 0.75, considered good to excellent in more than 80% of the items in most scales and Kappa was between 0.6 and 1.0 (good-excellent) for activities to which parents desired change in the home, school, and community settings.

On questions related to relevance, understandability, and time consumed for completing the PEM-CY, 94% (61/65) caregivers of children with disabilities reported that PEM-CY should be used in the intervention, 95% (62/65) found it relevant, 71% (46/65) found it easy to understand, and 58.4% (38/65) did not find it time-consuming to complete the measure. Twenty-seven percent (35/129) sought help for the participation sections and 39.5% (51/129) in the environmental sections. Thirty percent (27/129) participants needed assistance in both the participation and environment sections. Including the overlap between both these sections, a total of 57 percent (74/129) of caregivers needed assistance overall with various aspects of the instrument like with reading, understanding the items, and reflecting and correlating it with their personal experience.

## 4. Discussion

The ACEF framework provides a systematic method to culturally adapt and evaluate the PEM-CY in diverse contexts. Findings from our study suggest that the adapted version of the PEM-CY, modified based on in-depth interviews with caregivers, is a valid and reliable tool for assessing the participation of children and youth living in India.

The development of the original PEM-CY (content and layout) involved in-depth interviews and focus groups with parents/caregivers of children with and without disabilities living in North America, the end-users of these measures [9]. This has likely contributed to the widespread acceptability of the construct of participation across cultures and may explain why no modifications at the level of the “concept” were required. Caregivers of children with disabilities participating in our study too agreed that participation was a relevant construct and a measure like the Indian PEM-CY should be used in practice. Among certain cultures, parents associate successful participation with independence and autonomy in family life and the larger community [41]. Similarly, we observed that caregivers in our study initially tended to focus on the child’s abilities and independence while responding to questions related to the frequency and involvement or environment sections in various activities. In addition to this, having a deficit-based approach might be contributing to difficulties in understanding the contribution of the environment to a child’s ability to participate. However, the use of reminders and repeating instructions, as part of “operational” modifications, assisted in overcoming this challenge. In this sense, the PEM-CY provides a structured method and opportunity to discuss participation and factors that influence it thereby creating a shift towards a strengths-based, participation-focused approach to disabilities from a provider and caregiver perspective.

Further, we found that operational modifications were the most common in a low resource setting such as ours and studies conducted in other high resource contexts [17,42,43]. The need for operational modifications in high and low resource settings alike underscores the importance of health literacy in the caregiver respondents across settings [44]. Cognitive interviews were critical in establishing item, semantic, and operational equivalencies and identifying the need for a dual-mode of administration of the measure (self and provider administered).

Overall, the psychometric equivalencies including construct validity, test–retest reliability, and internal consistency of adapted PEM CY were adequate. The test–retest reliability was relatively lower on a few scales such as school frequency and community environment. This could be explained by the caregivers’ report on having less information and control over the child’s participation in school and some community settings during the cognitive interviews. As anticipated, this PEM-CY was able to identify the discrepancies between frequency and involvement in the participation of children with and without disabilities, supporting its construct validity. In addition, caregivers of children with disabilities desired change in participation more often and experienced more environmental barriers, as expected. While caregivers of children with disabilities perceived money, time, information, and supplies as being supports to a lesser extent than caregivers of children without disabilities these differences were not statistically significant. The physical, social, cognitive, or sensory demands of activity were not considered as supports by both groups. This is perhaps because caregivers in India are found to rely on informal social supports for enhancing the diversity of their child’s participation and to improve their well-being [13].

Participation is a relatively new construct for caregivers of children in the Indian context [14]. The use of the PEM-CY provides an opportunity to engage caregivers and children in identifying and utilizing formal and informal social supports that can improve participation. Such information can inform tailored intervention plans that address environmental barriers and supports identified by parents.

### Limitations and Future Directions

The diversity of the education level of caregivers participating in our study is not representative of the education levels of caregivers from rural India. Further, there are likely to be differences in environmental supports and barriers between urban and rural India. For these reasons, the Indian PEM-CY may be better suited for use in urban India. Caregivers who participated in our study endorsed the relevance, utility, and feasibility of the Indian PEM-CY. The PEM-CY has the potential to evaluate programmatic outcomes. Facilitators, barriers, and feasibility of such an exercise from a providers’ perspective needs to be examined in further studies.

## 5. Conclusions

The Indian PEM-CY is a reliable and valid measure that can be used in an urban context. The availability of a culturally adapted measure for evaluating the participation of children with disabilities offers a unique opportunity. At an individual level, it has the potential to reorient the child, caregiver, and health care provider focus on participation and environmental supports and barriers. Using parent interviews as a mode of administration offers service providers an important opportunity to dialogue with and influence the thinking about participation and the impact of the environment in resource-poor settings like ours.

## Figures and Tables

**Figure 1 ijerph-18-01514-f001:**
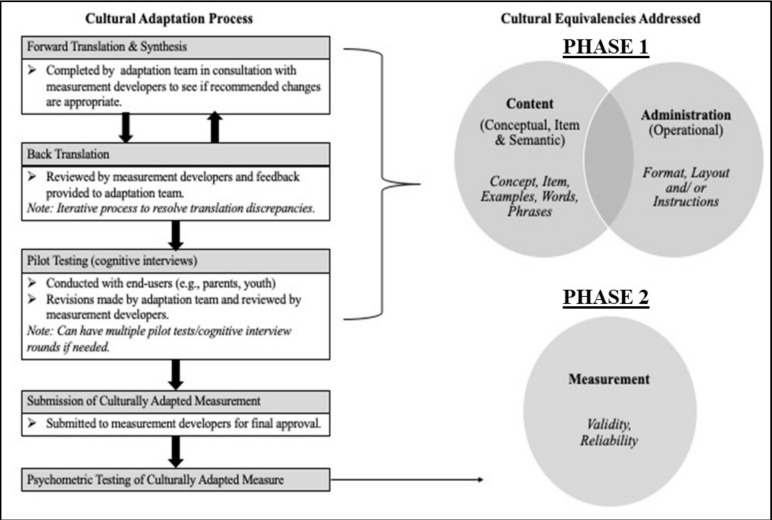
The adaptation process to achieve cultural equivalency for pediatric participation measures [20].

**Figure 2 ijerph-18-01514-f002:**
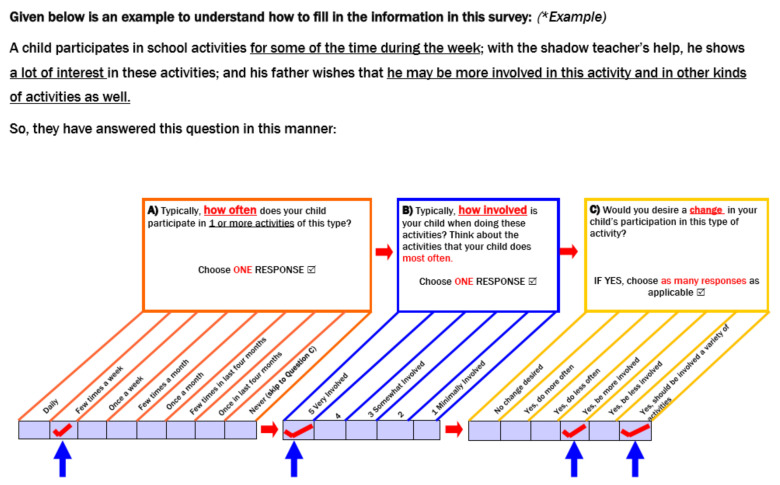
Clarifying survey instruction; (**A**), (**B**) and (**C**) subparts of the question on participation in a given setting elicit information about Frequency, Level of Involvement and Caregivers’ desire for change in frequency, involvement and variety of activities in which the child participates respectively.

**Table 1 ijerph-18-01514-t001:** Adapted version of Applied Cross-Cultural Equivalence Framework (ACEF) [17].

	Equivalence Criteria	Definition of Criteria
Phase1: Qualitative focus	Conceptual	The relevance of the underlying domain
	Item	Acceptability of items
	Semantic	Consistency of the meaning in the local language
	Operational	Suitability of instructions, administration, formatting, design
Phase 2: Quantitative focus	Measurement	Equivalence in the psychometric properties

**Table 2 ijerph-18-01514-t002:** Sociodemographic table of caregivers and children in Phase 1 (*n* = 15).

	Round 1 (*n* = 10)	Round 2 (*n* = 5)
Caregivers		
Fathers	1	1
Mothers	9	4
Caregiver education		
Up to High school	2	5
Graduation	6	0
Postgraduation	2	0
Employment status		
Employed	4	2
Unemployed	6	3
Monthly Family Income		
Below Minimum Wage (INR 10,000)	2	3
Above Minimum Wage (INR 10,000)	8	2
Children		
Sex		
Males	7	3
Females	3	2
Age		
5–8 years	7	1
8.1–12 years	2	4
12.1–17 years	1	0
Diagnosis		
Autism Spectrum Disorder	3	0
Autism Spectrum Disorder and Global Developmental Delay	1	0
Autism Spectrum Disorder and Intellectual Disability	1	0
Cerebral Palsy	1	0
Cerebral Palsy with Global Developmental Delay, Vision Impairment and Hearing Impairment	0	1
Global Developmental Delay	0	1
Global Developmental Delay and Learning Disability	1	0
Global Developmental Delay and Cerebral Palsy	1	0
Attention Deficit Hyperactivity Disorder	1	0
Language Disorder	1	0
Learning Disability	0	2
Intellectual Disability	0	1

**Table 3 ijerph-18-01514-t003:** Sociodemographic details of caregivers and children.

Variable	Cases (*n* = 65) *n*%		Controls (*n* = 65) *n*%	
Child Gender				
Male	37	57%	40	62%
Female	28	43%	25	38%
Child Age (Mean = 8.7 years)				
5–8	27	42%	27	41%
8.1–12	21	32%	21	32%
12.1–15	15	23%	13	20%
15.1 to 18	2	3%	4	6%
Autism Spectrum Disorder	21	32%	-	-
Specific Learning Disability	17	26%	-	-
Attention Deficit Hyperactivity Disorder	16	25%	-	-
Global Developmental Delay	16	25%	-	-
Child—Number of health conditions				
1	30	46%	-	-
2	11	17%	-	-
3	18	28%	-	-
0	6	9%	-	-
Child—Number of functional limitations				
1–3	16	25%	-	-
4–6	26	40%	-	-
7–9	19	29%	-	-
10–11	4	6%	-	-
Respondent relationship to the child				
Mother	49	75%	55	85%
Father	16	25%	6	9%
Other	0	0%	4	6%
Respondent age (years)				
18–29	3	5%	9	14%
30–39	36	55%	35	53%
40–55	24	37%	20	31%
Missing	2	3%	1	2%
Respondent education				
High School Education or lower	26	40%	26	40%
Graduate/Diploma/technical training	25	38%	27	42%
Postgraduate	14	22%	12	18%
Family income *				
Above Minimum Wage (INR 10,000/136.5 USD)	47	72%	50	76.9%
Below Minimum Wage (INR 10,000/136.5 USD)	17	26%	10	15.3%
Language of PEM-CY				
English	32	49%	37	57%
Hindi	33	51%	28	43%

* Minimum wages of Maharashtra State is 10,000 INR.

**Table 4 ijerph-18-01514-t004:** Group summary scores of children with (cases) and without disabilities (controls).

.			Cases (*n* = 65)		Controls (*n* = 65)				*n* = 130	*n* = 130
Home										
	Min	Max	Mean	Sd	Mean	Sd	Z/T value *	*p*-value	Internal consistency (Cronbach’s alpha)	Effect size (Cohen’s d)
Average frequency of home Participation	4.17	7	6.05	0.63	6.32	0.54	−2.649	0.008	0.7103	0.5759
Percentage of activities at home	10%	100%	87.23%	19.73%	92.92%	9.14%	−0.939	0.347		0.2419
Average of involvements at home	1	5	3.56	0.79	4.04	0.74	−3.716	<0.001	0.7158	0.7036
Home-percentage of change desired	0%	100%	77.28%	25.89%	54.91%	27.77%	−4.613	<<0.001	0.8161	0.8157
Home environment-score 3 (Support)	0	66.7%	27.692%	16.345%	32.692%	18.768%	−1.695	0.090	0.8303	1.2377 (HE)
Home environment-score 1 (Barriers)	0	66.6%	16.154%	16.853%	2.820%	6.633%	−5.837	<<0.001		
**School**										
	Min	Max	Mean	Sd	Mean	Sd	Z/T value *	*p*-value	Internal consistency (Cronbach’s alpha)	Effect size (Cohen’s d)
Average of school frequency **	1	7	4.73	1.32 (IQR: 1.70)	5.42	1.00 (IQR: 1.65)	−3.272	0.0013	0.6079	0.9225
Percentage of activities at school	20%	100%	68%	26%	83%	20%	−3.257	0.001		0.3038
Average of involvements at school	1	5	3.12	1.28	4.11	0.86	−4.457	<<0.001	0.7041	0.871
School percentage of change desired	0	100	78.65%	30.86%	45.47%	38.25%	−4.960	<<0.001	0.8535	1.0339
School environment-score 3 (Support)	0	94%	35.93%	21.57%	39.71%	18.98%	−1.195	0.232	0.8647	1.1139 (SE)
School environment-score 1 (Barriers)	0	70.5%	15.20%	19.28%	2.85%	5.55%	−4.618	<<0.001		
**Community**										
	Min	Max	Mean	Sd	Mean	Sd	Z/T value *	*p*-value	Internal consistency (Cronbach’s alpha)	Effect size (Cohen’s d)
Average of community frequency **	1.83	6.80	4.27	1.07 (IQR: 1.47)	4.81	1.17 (IQR: 1.44)	−2.724	0.0073	0.7355	0.8076
Percentage of activities at community	0	10%	54%	21%	66%	19%	−3.486	<0.001		0.0543
Average of involvements at community	1	5	3.13	1.12	3.87	0.91	−3.486	<0.001	0.7929	0.525
Community percentage of change desired	0	100%	73.18%	28.98%	49.74%	34.00%	−4.006	<<0.001	0.8626	0.7774
Community environment-score 3 (Support)	0	93.7%	28.17%	18.95%	30.19%	18.51%	−0.710	0.478	0.8768	1.0235 (CE)
Community environment-score 1 (Barriers)	0	81.25%	21.83%	22.78%	8.56%	13.05%	−3.876	<0.001		

* Z-value replaced with T-Value where an unpaired *t*-test was applied. ** Unpaired *t*-test was applied to the average school frequency and community frequency; IQR: Interquartile range; SD: Standard Deviation; HE: Home environment; SE: School environment; CE: Community environment; *p*-value of ≤ 0.05 was used as the cutoff for statistical significance.

## Data Availability

The data presented in this study are openly available in Fig-share at https://doi.org/10.6084/m9.figshare.13455110.v1.

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
