# Peer review of "Cross-Cultural Adaptation and Evaluation of the Participation and Environment Measure for Children and Youth to the Indian Context—A Mixed-Methods Study"

_ijerph, 2021, doi:10.3390/ijerph18041514_

Round 1
Reviewer 1 Report
The theme the manuscript addresses is highly relevant to enhance the participation of children with disabilities in countries all over the world and provides a substantiated measure to discuss participation of children with disabilities with different stakeholders, as caregivers, schools and communities to help them take their role in this regard.
I have a few comments/questions.
Line 153 and 154 state that test-retest reliability for all participation and environment sections were assessed over a 1-4-week period.
I wonder whether the variety in test-retest periods could be influencing the results. Also I wonder whether 1 week is not too short for a test-retest of this measure, because of memory effects.
Line 155 states "good discriminant validity", but does not give a value. I wonder why this value is not mentioned. If this value is available, it would be useful to mention it.
Line 307, table 3, mentions the number of child health conditions from 0 to 3. Often children with disabilities have several health conditions next to their primary diagnosis. I wonder what kind of health conditions are incorporated in this term? Only health conditions relevant to the diagnosis, or also other chronic health conditions, as asthma, COPD etc. May be it is useful to give some examples of conditions mentioned. In my experience children could well have more than 3 health conditions, so I wonder why in this table only 0 to 3 conditions are mentioned.
In line 342-344 it is stated that "Fifty-seven percent of caregivers (n=130) needed assistance with reading, understanding the items, and reflecting and correlating it with their personal experience. Twenty-seven percent sought help from the administrator in the participation sections and 39.5% in the environmental sections."
I wonder whether it should be made clearer in the discussion that given these percentages, the measure is only suitable to be administered by a professional in discussion with caregivers and not for self administration, to prevent misunderstanding.
Then a few minor textual suggestions:
Line 249: Hindi
Line 290-291 regarding the VAS -scales, it would help readability to add spaces between the VAS-values. Some are already in place, others are missing. If I understand correctly, the 3 numbers mentioned are related to understandability (1) and relevance of items (2) and relevance of examples (3). This is not instantly clear.
Line 314: at home, school and within the community
From line 345, table 4: I found it hard to read this table. I think better lay-out can help comprehension a lot. For example repeating the headers (min, max, mean, etc) for each setting, creating larger spaces between columns, etc.
Reviewer 2 Report
This is a well written paper. The application of the ACEP to the PEM CY is both novel and relevant. The authors described very succinctly the process and types of adaptations made and provided the results of this in a coherent manner. The importance of looking at measures for LMIC is argued and may be further supported by this reference.
Schlebusch, L., Huus, K., Samuels, A., Granlund, M., & Dada, S. (2020). Participation of young people with disabilities and/or chronic conditions in low‐and middle‐income countries: a scoping review. Developmental Medicine & Child Neurology, 62(11), 1259-1265.
The second phase on the psychometric properties are also clearly described and the results are presented clearly and coherently.
The approach to construct validity on page 9 line 309 - could be articulated more clearly. How was construct validity ascertained and what was the procedures/ analysis that were used. Some of these descriptions can be placed under data analysis as well.
